# Chemically modified graphene films with tunable negative Poisson's ratios

Yeye Wen [1,5], Enlai Gao [2,5], Zhenxing Hu[3,5], Tingge Xu[3], Hongbing Lu[3], Zhiping Xu[4] & Chun Li [1]

Graphene-derived macroscopic assemblies feature hierarchical nano- and microstructures that provide numerous routes for surface and interfacial functionalization achieving unconventional material properties. We report that the microstructural hierarchy of pristine chemically modified graphene films, featuring wrinkles, delamination of close-packed laminates, their ordered and disordered stacks, renders remarkable negative Poisson's ratios ranging from −0.25 to −0.55. The mechanism proposed is validated by the experimental characterization and theoretical analysis. Based on the understanding of microstructural origins, pre-strech is applied to endow chemically modified graphene films with controlled negative Poisson's ratios. Modulating the wavy textures of the inter-connected network of close-packed laminates in the chemically modified graphene films also yields finely-tuned negative Poisson's ratios. These findings offer the key insights into rational design of films constructed from two-dimensional materials with negative Poisson's ratios and mechanomutable performance.

[1] Department of Chemistry, MOE Key Laboratory of Bioorganic Phosphorus Chemistry & Chemical Biology, Tsinghua University, Beijing 100084, China. [2] Department of Engineering Mechanics, School of Civil Engineering, Wuhan University, Wuhan, Hubei 430072, China. [3] Department of Mechanical Engineering, The University of Texas at Dallas, 800 W. Campbell Rd., Richardson, TX 75080, USA. [4] Applied Mechanics Laboratory, Department of Engineering Mechanics and Center for Nano and Micro Mechanics, Tsinghua University, 100084 Beijing, China. [5] These authors contributed equally: Yeye Wen, Enlai Gao, Zhenxing Hu. Correspondence and requests for materials should be addressed to Z.X. (email: xuzp@tsinghua.edu.cn) or to C.L. (email: chunli@mail.tsinghua.edu.cn)

Macroscopic chemically modified graphene (CMG) films, processed from graphene oxide (GO) or reduced GO (rGO), feature inter-connected two-dimensional (2D) network structures, lightweight, exceptional mechanical properties, as well as high electrical and thermal conductivities[1–3]. These films have been demonstrated for applications ranging from membrane separation[4–8] to flexible energy storage devices[9,10]. Superior mechanical performance of the CMG films is a prerequisite for these applications. Recent studies on the mechanical properties of the CMG films are mainly focused on tensile strength, stiffness, and toughness, but few are on the Poisson's ratio, which is an important material parameter directly related to the modern functional materials[11,12].

The Poisson's ratio ($\nu$) characterizes the transverse deformation induced by uniaxial strain. For an isotropic, linear elastic and homogeneous material, thermodynamics requires $\nu$ to fall within $-1 < \nu < 0.5$[13], being from the most extendable to the most incompressible, although the limitation can be broken for anisotropic materials. Under uniaxial tension, most of the common materials contract in lateral directions to compensate for stretch-induced volume change, giving positive Poisson's ratios. Recent studies show that some natural and synthetic materials have a wide range of negative Poisson's ratios (NPRs) exhibiting auxetic behavior[11,12,14–17], which gives rise to such remarkable properties as enhanced toughness, indentation resistance, and shear stiffness. These enhanced mechanical properties allow the materials with notable NPRs to hold great promises for applications in aircraft, automotive, body-armor, and fiber composites with great pull-out resistance[11,12].

The Poisson's ratio of graphene monolayer has been characterized by molecular dynamics simulations[18–22]. Grima et al. discovered that the conformation of graphene can be modified via the introduction of defects (by removing atoms), and the crumpled conformation of graphene results in tunable NPRs, where a "crumpled paper" model was proposed to explain the NPR behavior of graphene[18]. It is also shown that the graphene monolayer exhibits large NPRs in specific directions if defects are introduced in a periodic arrangement, yielding a wavy form of graphene[22]. More recently, Wan et al. reported that the Poisson's ratios of GO monolayer can be effectively tuned by increasing the degree of oxidation in GO, reaching a value of $-0.57$ for fully oxidized graphene[20]. The dependence of the Poisson's ratio on the level of oxidation is attributed to the tension-induced suppression of ripples resulting from structural distortion, or in other works, through the disorder-related auxetic egg-rack mechanism. A considerable amount of effort has also been devoted to the Poisson's ratio in three-dimensional (3D), bulk graphene assemblies[23,24]. The results indicated that the microstructures of 3D CMG materials play a key role in modulating the Poisson's ratio. However, compared to the graphene and GO monolayers or their 3D bulk assemblies, little is known about the Poisson's effects in macroscopic CMG films.

In this work, we demonstrate that pristine GO and rGO films prepared by vacuum-assisted filtration and evaporation-induced self-assembly methods exhibit tunable NPRs in a wide range from $-0.25$ to $-0.55$ by engineering their chemistry and microstructures. Hierarchical structures of CMG films assembled from GO or rGO sheets are responsible for the NPR effect, as revealed by X-ray diffraction (XRD) and polarized Raman analysis. Theoretical model analysis clarifies that stretching induces suppression of the wrinkled close-packed laminates (CPLs), as well as their disordered stacks and delamination, which explains the tunability of NPR behavior in the CMG films. This well-controlled in-plane auxetic behavior, coupled with their lightweight and outstanding mechanical properties, makes CMG films promising for applications in aerospace, automotive, and defense.

## Results

**Preparation of CMG films**. GO sheets were synthesized by the modified Hummers method at oxidation temperatures 0 and 35 °C (labelled as $GO_0$ and $GO_{35}$, respectively, Supplementary Note 1)[25,26]. The structural characterization indicates a lower degree of oxidation, a larger graphitic domain, and fewer defects in $GO_0$ sheets compared to the $GO_{35}$ sheets (Supplementary Note 2 and Supplementary Fig. 1). Evaporation-induced assembly of concentrated GO dispersions ($5\,\mathrm{mg\,mL^{-1}}$) on a flat substrate or vacuum-assisted filtration of dilute GO dispersions ($1\,\mathrm{mg\,mL^{-1}}$) yields robust and uniform GO films, which could be easily peeled off from the substrate or membrane (Supplementary Fig. 2). All rGO films were obtained by post-reduction of relevant GO films with HI solution. For clarity, the abbreviation $x$-$GO_y$ is used to indicate the specimens, where $x$ represents the film-forming methods used ($e$ for evaporation and $f$ for filtration) and $y$ represents the oxidation temperature (0 for 0 °C with optimized intact structure[25] and 35 for 35 °C by the conventional procedure[26]). Specially, $x$-GO or $GO_y$ is used to represent GO samples with one specific characteristic. Thermal gravimetric analysis (TGA) shows that the solvent entrapped in the film does not depend on the fabrication methods. All CMG films feature smooth surfaces, laminated microstructures, and excellent mechanical properties as reported previously (Supplementary Note 3 and Supplementary Figs. 2 and 3)[1,3].

**Measurements of Poisson's ratios**. A Poisson's ratio is defined as the negative ratio of the transverse strain to the longitudinal strain in the loading direction under uniaxial tension. Thus, accurate determination of the local transverse strain within the region of uniform deformation and avoiding gripping effect in measuring longitudinal strain of a specimen proposed in our previous work[27] are prerequisites for evaluating the Poisson's ratio accurately, and the measurement of Poisson's ratio on a film specimen remains challenging. Herein, fluorescent digital image correlation (FDIC) is used to measure the in-plane deformation by tracking and matching a distinct speckle pattern in the reference and the deformed images during the loading process[28,29]. This approach allows measurement of the full-field displacements with sub-pixel accuracy. Fluorescent particles are used to form speckle patterns to keep track of movement of material points in the film under deformation. This approach eliminates the light specular reflection (Fig. 1a). Typical FDIC speckle images of $e$-$GO_0$ film are shown in Fig. 1b–e, where $X$ and $Y$ are the transverse and loading directions, $U$ and $V$ are the corresponding displacement fields along the $X$ and $Y$ directions, respectively. The results show that the strains along the transverse and loading directions ($\varepsilon_{xx}$ and $\varepsilon_{yy}$) of the GO film are both positive, and the average transverse strain increases with the increase of average longitudinal strain, signaling a NPR behavior. Based on the transverse strain $\varepsilon_X$ and longitudinal strain $\varepsilon_Y$ averaged from the strain fields $\varepsilon_{xx}$ and $\varepsilon_{yy}$ (Fig. 2a and Supplementary Fig. 4), the instantaneous in-plane Poisson's ratio is calculated as:

$$\nu_{YX} = -\mathrm{d}\varepsilon_X/\mathrm{d}\varepsilon_Y \tag{1}$$

As shown in Fig. 2a, upon uniaxial loading along the $Y$ direction for $e$-$GO_0$ film, the $\varepsilon_X$ increases linearly with $\varepsilon_Y$, giving an instantaneous Poisson's ratio around a constant value of $-0.47$, which is independent of the applied longitudinal strain (Fig. 2b). Impressively, it is found that all the examined GO and rGO films exhibit NPR behavior (Supplementary Fig. 4), irrespective of the GO precursors used, film-forming methods, and the chemical structure of CMG (GO or rGO) sheets. The

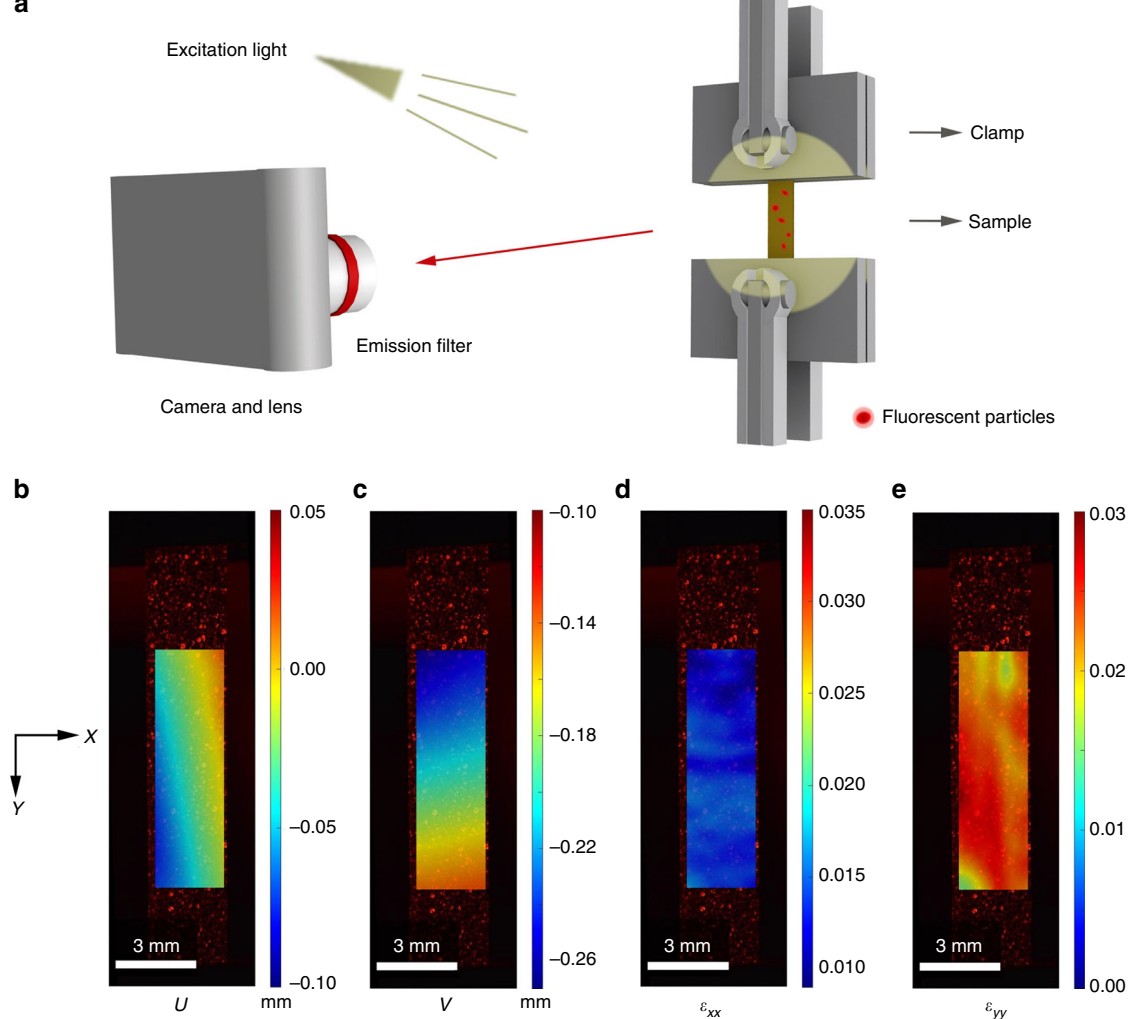

**Fig. 1** Tensile tests of chemically modified graphene films monitored by fluorescent digital image correlation. **a** Schematic diagram of the fluorescent digital image correlation (FDIC) measurement, where camera and lens detect light emitted from red fluorescent particles deposited onto the $e$-$GO_0$ films (prepared by evaporation of GO solution synthesized at 0 °C). The emission filter allows only light from the excited fluorescent particles to pass. **b**–**e** Optical images of $e$-$GO_0$ films with fluorescent particles on its surface, and the displacement fields $U$ (**b**) and $V$ (**c**), and the strain fields measured in the transverse (**d**) and longitudinal (**e**) directions from FDIC analysis. Source data are provided as a Source Data file

average Poisson's ratios for the films are determined as −0.26 ($e$-$GO_{35}$), −0.25 ($f$-$GO_{35}$), −0.44 ($f$-$GO_0$), −0.48 ($f$-$rGO_0$), −0.53 ($e$-$GO_0$), and −0.55 ($e$-$rGO_0$) (Fig. 2c), via Eq. (1) using the full range of data (Supplementary Fig. 4). Moreover, cyclic loads with a peak strain of 1.5% does not induce significant changes in the Poisson's ratio even after 15 cycles (Fig. 2d and Supplementary Fig. 5), indicating that the NPR effect is inherent for the CMG films.

**Microstructural basis of the negative Poisson's effects**. The chemical structures of GO building blocks modulate the microstructures and Poisson's ratio of the GO films. As shown in Fig. 3a, all XRD patterns of $GO_0$ and $GO_{35}$ films display sharp diffraction peaks, suggesting the formation of CPLs, which is evident from the scanning electron microscopy (SEM) images of CMG films (Supplementary Fig. 2b)[1]. The interlayer spacings of $e$-$GO_0$ and $f$-$GO_0$ films are calculated as 0.806 and 0.785 nm, respectively, based on the diffraction peaks at $2\theta = 10.95°$ and 11.25° with the full-width at half-maximum (FWHM) of 1.23° and 1.05°. $e$-$GO_{35}$ and $f$-$GO_{35}$ films feature smaller interlayer spacings of 0.772 and 0.750 nm, respectively, which are

determined from diffraction peaks at higher angles $2\theta = 11.44°$ and 11.77° with narrower FWHMs (0.56° and 0.47°). The XRD data can be used to evaluate the degree of disorder of GO sheets in the CMG films, where the finite FWHM indicates the sheet mis-alignment[30,31]. It is distinct that the $GO_0$ films with larger FWHM have larger absolute values of NPRs (−0.53 ($e$-$GO_0$) and −0.44 ($f$-$GO_0$)) than $GO_{35}$ films (−0.26 ($e$-$GO_{35}$) and −0.25 ($f$-$GO_{35}$)). The increased amount of organosulfate (3.33 at% sulfur) in $GO_0$ films relative to that (0.77 at%) in $GO_{35}$ films is suggested to account for the larger $d$-spacing and less ordered alignment of $GO_0$ films[26,32]. The interlayer spacing between GO sheets characterized by XRD measures the packing density in CPLs, which is high enough to suppress the out-of-plane corrugation at the sheet level, which was proposed to be responsible for the NPR effects of graphene and GO monolayers[18–22].

The structural differences between the GO sheets prepared by different synthetic protocols are further identified by UV–vis absorption spectra. The main absorption peak of GO around 230 nm is associated with the $sp^2$ domain in GO sheets. The absorption peak of $GO_0$ (236 nm) is red-shifted with respect to that of $GO_{35}$ (231 nm), demonstrating that $GO_0$ sheets have a larger graphitic domain than that in $GO_{35}$, since the carbon atom

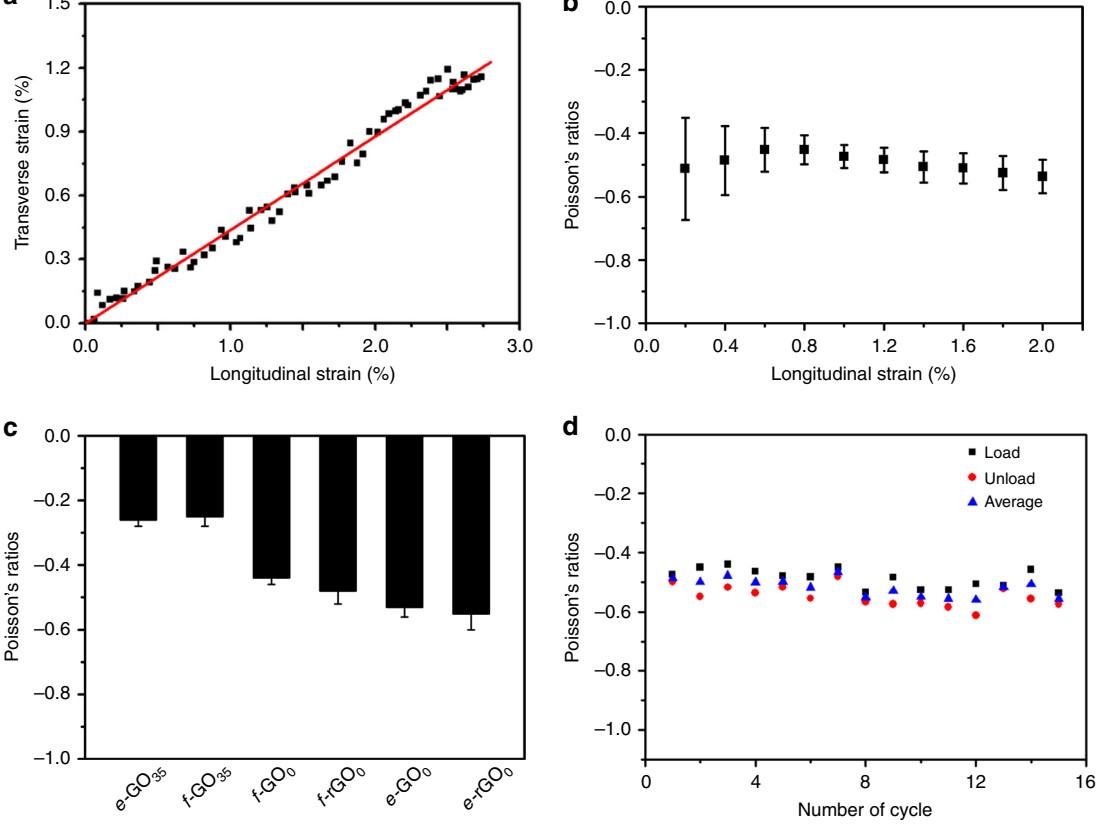

**Fig. 2** Typical mechanical response of chemically modified graphene films. **a** The transverse strain $\varepsilon_X$ plotted as a function of the longitudinal strain $\varepsilon_Y$ of $e$-$GO_0$ films under a tensile loading in the $Y$ direction. The red line is the linear fitting line. **b** The instantaneous Poisson's ratios of $e$-$GO_0$ films averaged from three samples in Fig. S4b. **c** Poisson's ratios of $e$-$GO_{35}$ films (prepared by evaporation of GO solution synthesized at 35 °C), $f$-$GO_{35}$ (prepared by filtration of GO solution synthesized at 35 °C), $f$-$GO_0$ (prepared by filtration of GO solution synthesized at 0 °C), $f$-$rGO_0$ (reduced $f$-$GO_0$ films), $e$-$GO_0$ and $e$-$rGO_0$ (reduced $e$-$GO_0$ films). **d** Poisson's ratios of $e$-$GO_0$ films under cyclic uniaxial tensile loading with a peak strain of 1.5%. The error bars in panels (**b**, **c**) indicate the standard errors (three samples were measured). Source data are provided as a Source Data file

rupture is prevented to some extent by a relatively lower temperature of oxidization (Supplementary Fig. 6)[33,34]. The low defect density and larger $sp^2$ domains within $GO_0$ basal planes facilitate the van der Waals interactions between neighboring $GO_0$ sheets[34,35]. Consequently, in film-forming processes by evaporation-induced assembly or vacuum-assisted filtration, the mobility of $GO_0$ sheets in the dispersions is limited, and it becomes difficult to adjust the sheet conformation to adopt an energetically-favorable, ordered alignment, resulting in the formation of $GO_0$ films, where the hierarchical structures of GO sheet-containing CPLs is less ordered compared with $GO_{35}$ films.

The Poisson's ratio of the CMG films depends on the film microstructures tailored by the film-forming methods. As seen in Fig. 3a, with the same GO precursors, evaporation-induced assembly of concentrated GO dispersion (5 mg mL$^{-1}$) produces $e$-GO films with larger interlayer spacing and FWHM, together with more significant NPR effects than the $f$-GO films fabricated by vacuum-assisted filtration from dilute GO dispersions (1 mg mL$^{-1}$). GO sheets intend to form a low-density dynamic network at a relatively higher concentration ($\geq$3 mg mL$^{-1}$) due to the presence of strong physical cross-linking sites[26,36], leading to the formation of CMG films with more wrinkled textures of the CPLs. Considering all of these factors, we conclude that the Poisson's ratio of CMG films is highly related to the topological structure of CPLs, which can be modulated by the chemical structures of GO building blocks and the film-forming methods. The wrinkled textures of CPLs enhance the NPR effect of the CMG films. After

reduction with hydroiodic acid solution, the rGO films feature similar hierarchical structures with wrinkled textures and the associated auxetic behavior exhibited in the GO films (Fig. 2c and Supplementary Fig. 2).

To gain further insights into the NPR behavior of GO films, $e$-$GO_0$ films with different orders of alignment of CPLs were prepared by pre-stretching the wet GO films during the film-forming process. It is found that upon increasing the load, the resultant $e$-$GO_0$ films exhibit smaller absolute values of NPRs and narrower FWHMs, ranging from $\nu = -0.53$ (FWHM = 1.23°, without pre-stretch) to $\nu = -0.46$ (FWHM = 1.13°, with 30 mN pre-stretch), and $\nu = -0.31$ (FWHM = 1.04°, with 60 mN pre-stretch) (Fig. 3b and Supplementary Figs. 7 and 8). Consequently, pre-stretch induces conformation changes of the CPLs in the CMG films, which are gradually aligned to a more energetically-favorable arrangement at the expense of forming wrinkled textures. These results demonstrate that the NPR behavior of the CMG films can be readily tuned by controlling the topological structure of CPLs and their stacks in the films. This observation has also been confirmed by polarized Raman spectral examinations of CMG films prepared under different amplitudes of pre-stretch.

Polarized Raman spectra were recorded by directing an incident laser beam parallel to the base plane of $e$-$GO_0$ films with pre-stretch. As shown in Fig. 3c, $\alpha$ is the average angle of alignment between the CPLs and the loading direction, and $\beta$ is the angle between the electric field vector of the incident laser and the base plane of the $e$-$GO_0$ film, which is tuned from 0° to 360°

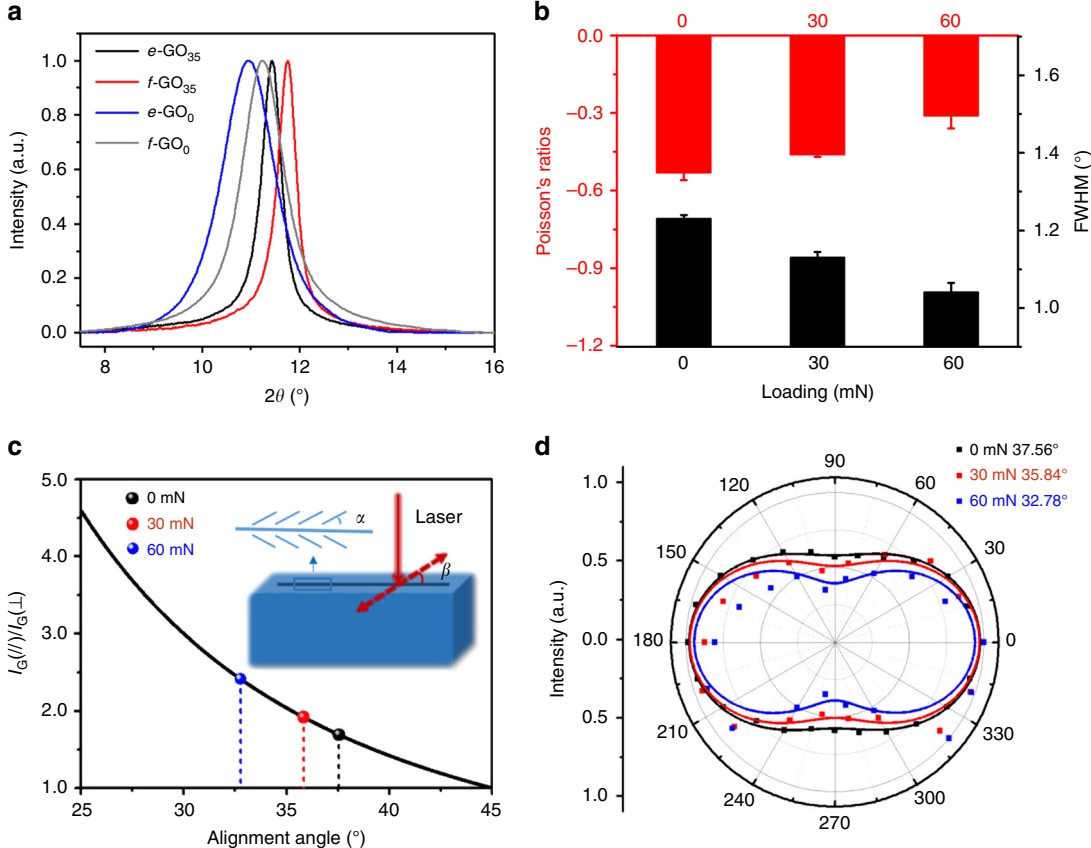

**Fig. 3** Microstructural origins of the negative Poisson's ratio (NPR) effect. **a** X-ray diffraction patterns of the e-GO$_{35}$, f-GO$_{35}$, e-GO$_0$, and f-GO$_0$ films. **b** Poisson's ratios and the full-width at half-maximum (FWHM) measured for the e-GO$_0$ films under different amplitudes of pre-stretch. **c** Relationship between the value of $I_G(//)/I_G(\perp)$ and the angle of sheet ($\alpha$) in the films (inset: the polarized Raman setup). **d** The G band intensity, $I_G(\beta)$, plotted as a function of $\beta$. Black dots and lines are the experimental data of e-GO$_0$ films without pre-stretch and the fitting curve using Eq. (2), while the red (blue) ones are the data and fitting curve for e-GO$_0$ films under 30 (60) mN pre-stretch. The error bars in panel (**b**) indicate the standard errors (three samples for the Poisson's ratios and four samples for FWHM). Source data are provided as a Source Data file

(Supplementary Fig. 9). The G band intensity in polarized Raman spectra of the CMG films is plotted against the angles $\alpha$ and $\beta$ in Fig. 3d, which follow the equation:

$$I_G = \frac{c^2}{2} \cos^2\alpha \{2 + \cos[2(\alpha - \beta)] + \cos[2(\alpha + \beta)]\} \quad (2)$$

For samples with the angle of alignment $\alpha$, the value of $I_G(\beta)$ characterized experimentally reaches a maximum at $\beta = 0°$ and 180° ($I_G(//)$), whereas $I_G(\beta)$ reaches a minimum at $\beta = 90°$ and 270° ($I_G(\perp)$)[37]. The ratio $I_G(//)/I_G(\perp)$ equals $\cot^2\alpha$ (Fig. 3c). Fitting the data of normalized Raman G band intensity with the value of $\beta$ between 0° and 360° using Eq. (2), one obtains the value of $\alpha$ decreasing from 37.56° to 35.84°, and 32.78° upon enhancing pre-stretch (Fig. 3c, d). Based on these results, it becomes clear that the NPR of CMG films can be tuned by modifying their hierarchical microstructures, where the interconnected network of CPLs define the NPR behavior of the CMG films.

**Theoretical analysis**. To understand the microstructural evolution of CMG films under tensile loading and quantifying the NPR behavior, we constructed a multiscale model for the CMG films based on experimental evidence. Specifically, based on SEM and optical images of the cross-section and surface topography of the CMG films, respectively (Fig. 4a, b), we illustrated the microstructures of monolithic CMG films in Fig. 4c, which include inter-connected CPLs (or GO multilayers with typical thickness of

~10–40 nm). Their evolution under tensile loading involves aligning of disordered stacks of CPLs, reduction in the out-of-plane corrugation of wrinkled CPLs and their delamination. These processes are activated along the tensile direction, but the misalignment in other directions is also suppressed, as a consequence, endowing the CMG films with NPRs (Fig. 4c). The evidence of these processes can be found from the polarized Raman data. Compared with the flattening process of a GO or graphene monolayer[20], the out-of-plane corrugation of the inter-connected network of CPLs in the CMG film is stabilized by its microstructural hierarchy, and can have much higher amplitude, leading to more prominent NPR effects as we identified experimentally. With these arguments, we rationalize the mechanisms through a representative volume element (RVE) featuring wavy textures of CPLs (Fig. 4d), following the spirit of the model of foldable structures proposed by Grima et al.[38] and the cubic lattice model coined by Baughman and co-workers[39]. In our model, the sticks (solid and dash lines in Fig. 4d) represent the out-of-plane corrugation of CPLs resulted from disordered stacks, wrinkles, or delamination. The angle $\theta$ between the sticks and the basal plane of a CMG film measures the out-of-plane corrugation of CPLs, and is related to the alignment angle $\alpha$ measured in polarized Raman experiments (see Supplementary Note 4 for details). The $\nu$–$\alpha$ relation predicted from our model aligns well with experimental data (Fig. 4e), suggesting that the NPR effect of CMG films does originate from the wavy texture of the inter-connected CPL network and will be reduced if the alignment of CPLs improves.

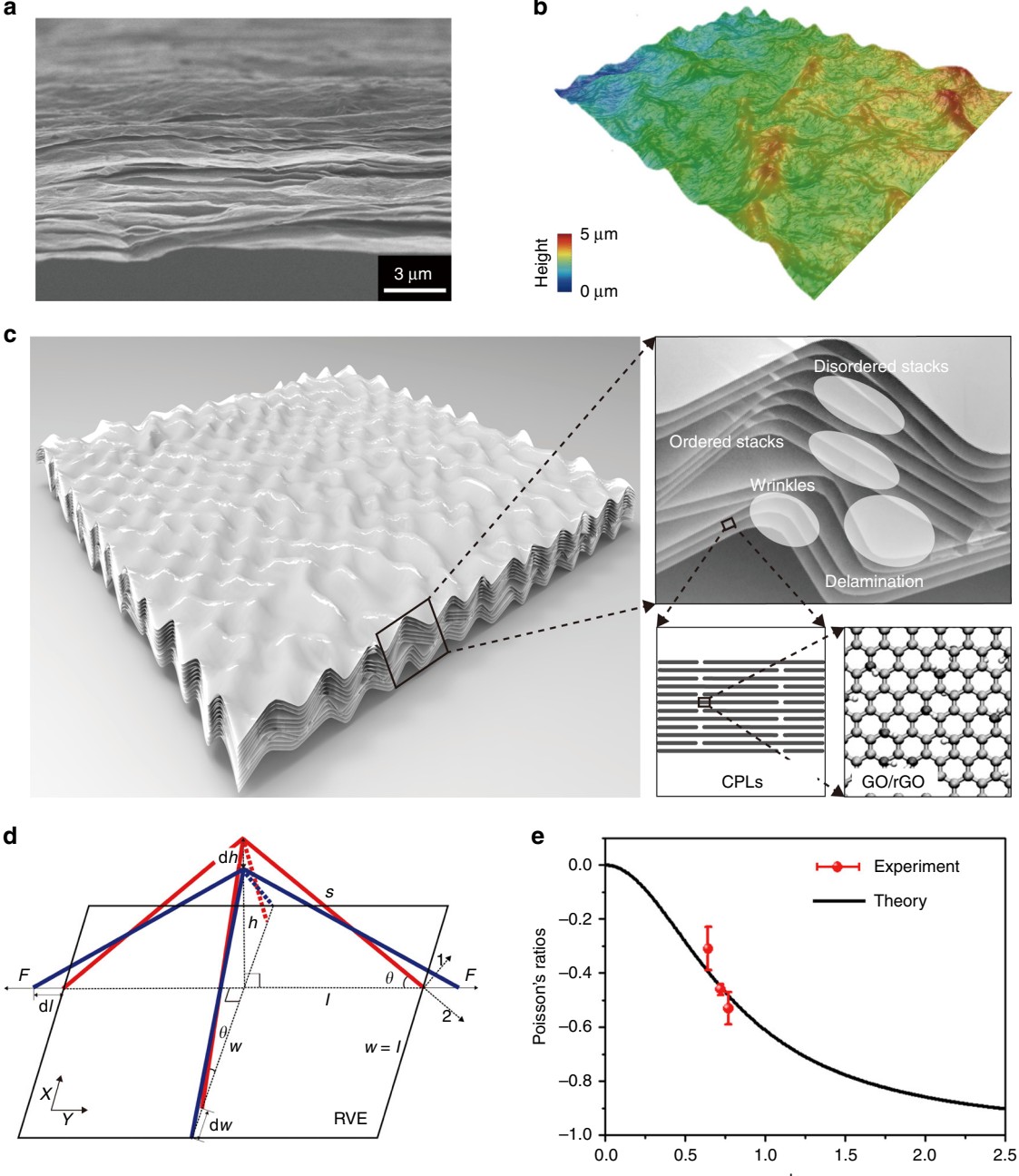

**Fig. 4** Mechanisms of negative Poisson ratio (NPR) behaviors explained by the theoretical model. **a** Wrinkles, delamination of close-packed laminates (CPLs) and their ordered, disordered stacks in e-GO$_O$ films characterized by scanning electron microscopy (SEM). **b** Three-dimensional reconstruction of the surface topography of e-GO$_O$ films through optical microscopy. **c** Illustration of the multiscale hierarchical microstructure of a chemically modified graphene (CMG) film featuring individual graphene sheets as the fundamental building blocks, CPLs and their inter-connected network. **d** A 4-stick representative volume element (RVE) model, where the red configuration is stretched by force $F$ to the blue one. **e** Theoretical predictions and experimental measurements of the Poisson's ratio as a function of the angle of alignment between CPLs, $\alpha$, which is related to the value of $\theta$ in the model (see Supplementary Note 4 for details). The error bars for experimental data in panel (**e**) indicate the standard errors (three samples were measured). Source data are provided as a Source Data file

## Discussion

We have demonstrated that the CMG films fabricated by solution-processed approaches feature NPRs with the value of $\nu$ ranging from −0.25 to −0.55. It should be noted that the in-plane and out-of-plane microstructures of CMG films are strongly anisotropic. The aligning process of CPLs only results in in-plane NPR effects, while the Poisson's ratio measured in the out-of-plane direction is always positive, with a reduction in the thickness of CMG films by ~20% at a tensile strain of 3.5% as reported in our previous work[27].

Our hypothesis on the microstructural evolution in CMG films is limited by the resolution of structural characterization techniques. Relative sliding between CPLs is not considered, which may weaken the suppression of out-of-plane corrugations of CPLs under tensile loading, and thus reduce the NPR effect. The morphological changes of GO sheets in the CPLs are expected to be strongly constrained by the layer-by-layer packing. The chemical structures of GO sheets are thus correlated to the NPR behaviors through their modulation in the film-forming processes.

Future work on the characteristic microstructural features and their statistics from experimental measurements is required to extract the key parameters for constructing numerical models, which can extend the current analysis by considering more realistic models. Our RVE-based theory neglects the wide distribution of the microstructural features resulted from the coarse control of film preparation processes, but captures the most essential mechanism of the NPR effect by introducing the concept of CPLs. Our experimental characterization and analysis indicate that the wavy textures of close-packed GO laminates as the building blocks are responsible for the observed auxetic behavior in the macroscopic GO and rGO films. The argument is validated by further engineering control of the NPR behaviors. The wrinkled texture of the inter-connected CPL network within the films is readily tailored by the chemical structures of the GO precursors and the film-forming methods, resulting in films with tunable NPR values.

The microstructural hierarchy of CMG films featuring graphene sheets as the building blocks, the CPLs and their inter-connected networks is reminiscent of the three-level model of carbon nanotube fibers (individual nanotubes as building blocks, the close-packed bundles, and their inter-connected network)[40]. These abstracted concepts help to improve understanding of the multiscale nature of nanostructured assemblies, their evolution under mechanical loading, and their impacts on the macroscopic mechanical performance[41]. The current study thus furnishes design guidelines for the construction of films with NPR behaviors from 2D building blocks for exhibiting auxetics.

## Methods

**Preparation of the GO and rGO films**. Evaporation-induced assembly of $GO_0$ or $GO_{35}$ dispersions on a polystyrene Petri dish under ambient condition yield $e$-GO films, which can be easily peeled off from the substrate for further characterization. The evaporation process typically takes 3 days at 25 °C. $f$-GO films were fabricated by vacuum-assisted filtration of dilute GO dispersion (1 mg mL$^{-1}$) through a poly-(tetrafluoride ethylene) (PTFE) membrane (47 mm in diameter, 0.22 μm pore size). All rGO films are prepared by chemical reduction of the relevant GO films with a mixed ethanol/aqueous 57% HI solution (3/1 by volume) at room temperature for 12 h. The resulting rGO films were repeatedly washed by ample ethanol and dried at room temperature for 24 h before characterizations. Pre-stretched GO films were prepared by sandwiching GO rectangular strips (3 mm × 30 mm) in a home-made apparatus with a given loading under 100% relative humidity (RH) for 1 week, followed by air-drying the resultant GO films under 50% RH for 1 day. The GO strips were fabricated by razor blade cutting wet GO films, which were fabricated by cast-drying GO dispersions (5 mg mL$^{-1}$) on PTFE Petri dish under 50% RH for 2 days.

**Microscopic observations**. SEM images were recorded using a Sirion 200 field emission SEM. The samples used for size distribution statistics by SEM were fabricated by deposition of diluted GO solution (5–10 μg mL$^{-1}$) on a 300-nm-thick-SiO$_2$/Si wafer followed by drying in a clean bench without sputter-coating any conductive layers. Surface morphology of $e$-GO$_0$ films was recorded and reconstructed using a 3D optical microscope (KH-8700 HIROX, 2500X).

**Structure characterizations**. X-ray photoelectron spectroscopy (XPS) data was recorded on an ESCALAB 250 photoelectron spectrometer (ThermoFisher Scientific) with Al Kα (1486.6 eV) radiation. XRD patterns of the films were collected on D8 Advanced X-ray diffractometer with Cu Kα radiation ($\lambda = 0.15418$ nm, Bruker, Germany) with scanning speed of 5° per minute at room temperature. Raman spectra were collected on LabRAMHR instrument (Horiba Jobin Yvon) with a 532-nm wavelength laser. During the polarized Raman characterization of CMG films, the polarization state of laser was tuned by the half-wave plate positioned in the beam path, and a man-made stage was used to vary the angle of film according to the original direction of linearly polarized excitation.

**Mechanical characterizations**. All samples for tensile testing were cut into rectangular strips with width of 3 mm and length of 30 mm by a single-edge razor blade and attached to a frame of card paper by epoxy glue. Uniaxial tension of the GO and rGO films was carried out using an Instron 3342 universal testing machine (Instron, USA). The transverse strain was determined using an optical microscope to record the videos of the fluorescent particle movement on the film surface at a strain rate of $5 \times 10^{-4}$ s$^{-1}$ until the film broke. FDIC was used to track and match

the fluorescent patterns in the reference and deformed states. Red fluorescent particles (580/605, Life Technology Corp., #F13083) were used in the fluorescent microscope, with excitation and emission/barrier filters for fluorescent imaging. The primary function of the emission/barrier filter in the fluorescent microscope is to block the excitation wavelengths used and allow only the excited light from the fluorescent particles to pass. Matching filters were thus used for fluorescent imaging (ET560/40x and ET615/40m, Chroma Technology Corp.). The samples were randomly sprayed with fluorescent particles using an airbrush. It should be noted that the displacement measurement in FDIC is sensitive to the sample alignment, and errors may arise from sample gripping. However, for our FDIC data, the angle of misalignment is ~5°, corresponding to an error in the strain of ~0.001. The FDIC-calculated strain is an order of magnitude higher than this level of error, and thus the strain measurement is reliable.

## Data availability

The data sets within the article and Supplementary Information of the current study are available from the authors upon request. The source data underlying Figs. 2, 3, and 4e and Supplementary Figs. 1b, c, e, f, 3–9 are provided as a Source Data file.

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

## Acknowledgements

Y.W. and C.L. are supported by National Natural Science Foundation of China (51673108 and 51433005) and National Key R&D Program of China (2016YFA0200202). Z.X. acknowledges National Natural Science Foundation of China through Grants 11825203 and 11472150. E.G. acknowledges the support from the starting-up fund of Wuhan University and the Fundamental Research Funds for the Central Universities. Z.H., T.X. and H.L. are supported through Grants NSF CMMI-1636306, CMMI-1661246, and CMMI-1726435, and Louis A. Beecherl Jr. Chair. The authors also thank Prof. Gaoquan Shi for discussion and his valuable advices on this work.

## Author contributions

C.L. and Z.X. conceived the idea and directed the project. Y.W. carried out materials synthesis, performed materials characterization, and related data analysis. Z.H., T.X. and H.L. offered FDIC methods and equipment for the mechanical tests. Y.W., E.G. and Z.H. measured the Poisson's ratio. Y.W. and E.G. analyzed the data of Poisson's ratios. E.G. developed the theoretical model. All authors participated in the interpretation of the data and the writing of the manuscript.

## Additional information

**Competing interests:** The authors declare no competing interests.

