## [Peer Review File · Nature Communications]

Reviewers' comments:

Reviewer #1 (Remarks to the Author):

The manuscript of Wen et al. presents experimental evidence that appropriately modified graphene can be made to exhibit negative Poisson's ratio. This finding is important and timely and appears to be scientifically sound and thus merits report in the primary literature. Some minor comments:

(1) The range $-1 < \nu < +0.5$ for isotropic materials is incorrect as the 'smaller than' sign ($<$) should be replaced by smaller or equal to. Furthermore, given that material which is being studied is obviously not isotropic as its out of plane properties are likely to be very different from what is observed in plane, these limits do not apply. This should be made clear.

(2) Poisson's ratio is in general not isotropic and hence it is important that when reporting it one also indicates the place where it is being measured and the direction of loading. Thus in this case, the symbol ν in equation (1) should be replaced by ν_{yx} to indicate that the measurement is being done in the xy -plane for stretching in the y -dimension.

(3) In the measurement of the Poisson's ratio, the same result could probably have been obtained from data of the measurements of dimensions (say 'X' and 'Y') without introducing stress since the instantaneous Poisson's ratio $\nu_{yx} = -(dX/X)/(dY/Y) = -(dX/dY)*(Y/X)$. The reviewer prefers if this form of the equation is also included and the authors check whether the data can be manipulated in such a way that the Poisson's ratio is calculating without invoking a third parameter, in this case stress.

(4) The reviewer is pleased to note that finding of this work is confirming earlier work by Grima et al. (ref. 17) which had likened wrinkled graphene films to a macroscopic 'crumpled paper' model. This should be explicitly stated.

(5) Can the authors comment on isotropy / anisotropy in the Poisson's ratio? Here, the Poisson's ratio is measured for stretching in a particular direction. Whilst it may be inferred that the negative in-plane Poisson's ratio is likely to be accompanied by a large positive Poisson's ratio as the sheet is becoming increasingly more flat as it is stretched, it may equally be extrapolated that the in-plane Poisson's ratio measured for stretching in the Y direction could have been measured for stretching in any other in-plane direction. The authors should also note that since the original Grima et al. paper (ref. 17), there was also another paper which showed that graphene could be made to exhibit even

higher negative Poisson's ratio in specific directions if the defects are placed in a periodic manner to get a 'wavy' form of graphene (see Grima et al. (2018) Ann der Physy.). This latter paper should also be cited.

(6) The authors should specifically that it is 'shape' of the sheet which is giving rise to the specific Poisson's ratio properties, not the actual chemical composition. The authors should check their manuscript well to ensure that this is clearly reflected. In particular, the sentence on the few lines of Page 7 should be re-phrased.

(7) The reviewer is of the opinion that the mathematical model present (which is a variation of an earlier model presented in 2011 by Grima et al., DOI 10.1002/pssb.201083982) should be presented in the SI. The main text should just give a summary of the model, together with the appropriate reference to earlier work, which apart from this 2011 paper should also cite a much earlier 1998 paper 'Negative Poisson's ratios as a common feature of cubic metals' by RH Baughman et al.

(8) The sentence ... indicating that the NPR, i.e., auxetics, is an intrinsic property of CMG films ... should be rephrased ... 'auxetics' should be 'auxeticity' and the word 'intrinsic' is not the best word to use.

(9) Towards the end of the Discussion section, within the sentence starting: The findings provide an understanding of the performance... 'additiona' should be 'addition'.

Reviewer #2 (Remarks to the Author):

In this paper, the authors reported a wide range of tunable negative Poisson's ratio in CMG membranes in experiments. They found that different synthesis procedures always led to a NPR. They attributed the measured NPR to the intrinsic wrinkles and disordered stacks in CMG films. Via controlling synthesis procedures (such as pre-stretching) to tune the microstructure hierarchy, they can obtain different NPRs in experiments. This feature is very important in practical applications. NPR is an interesting property that has stimulated intensive research interests recently. It has great promises in many applications ranging from aircraft, defence, to biomedical technologies. The authors also proposed a mechanical model that can quantitatively predict the NPR, which agrees with experimental results well. This model will be useful for design. This is an interesting work. This referee has some technical issues and requests the authors to address.

(1) The authors showed that the pre-stretching during sample synthesis is an effective way to change NPR. But they did not provide experimental details of how to apply the pre-stretching. This information could be useful for others to further develop this method for real applications.

(2) Could the authors explain how they estimated the quantity α in Eq. (6) from experimental measurements? This quantity is critical in their model.

(3) In Fig. S3, the CMG films have nonlinear elasticity. Does this have any correlation with the variation of the instantaneous Poisson ratio as shown in 2b?

(4) They have used symbol α in Eq. (2) and (6). But apparently, they have a different meaning in these two equations.

(5) In Figure 1b, the U displacement field is somehow not aligned with the loading direction. Could the authors provide some explanation? Will this affect the calculated strain value?

(6) In page 3, 2nd paragraph, 4th line, the authors stated 'Under an uniaxial tension, all monolithic and homogeneous ...'. Did they refer to isotropic materials?

(7) page 3, first paragraph, line 7, 'stifness' typo?

Reviewer #3 (Remarks to the Author):

Wen et al. present a study on Chemically Modified Graphene Films with Tunable Negative

Poisson's Ratios. The manuscript is well written but I believe the conclusion are not adequately supported by the data presented and publication in any form would be premature.

1. It is unclear how the authors assume monolithic properties for a GO or rGO microstructure with several nanoscopic platelets stacked. The underlying theoretical assumptions should be clearly explained to place this study on a solid theoretical footing.

2. The authors should explain if there is any relative movement of the flakes in the microstructures as a function of loading? How does this affect the Poisson ratios computed?

3. The authors should explain if there is solvent entrapment within the microstructures that gives rise to differences seen for evaporated vs vacuum filtered and 0 vs 35 fabrication methods.

4. Raman and UV-Vis are microscopic probes and it is unclear how the macroscopic properties (Poisson ratio) of the microstructure are directly related to microscopic properties.

5. How controllable and re-producible is the Poisson ratio for each method of fabrication? Why is the variability largest for the evaporated sample compared to vacuum filtered?

6. The authors attribute much of the observed properties to wrinkles and crumples in GO or rGO but present no direct evidence that is conclusive/unambiguous. It is hence unclear what is the fundamental mechanisms by which the microstructures exhibit negative Poisson's ratio?

7. How do the authors rule out delamination of the films comprising the microstructure leading to the observed negative Poisson ratio?

Reviewer #1:

The manuscript of Wen et al. presents experimental evidence that appropriately modified graphene can be made to exhibit negative Poisson's ratio. This finding is important and timely and appears to be scientifically sound and thus merits report in the primary literature.

Reply: Thank you for your recommendation. We polished our manuscript following the suggestions from you and other reviewers. Please find below a detailed, point-to-point responses to your minor comments.

Some minor comments:

(1) The range $-1 < \nu < +0.5$ for isotropic materials is incorrect as the 'smaller than' sign ($<$) should be replaced by smaller or equal to. Furthermore, given that material which is being studied is obviously not isotropic as its out of plane properties are likely to be very different from what is observed in plane, these limits do not apply. This should be made clear.

Reply: We corrected the expression and added notes on the applicability in anisotropic materials explicitly.

Changes made:

In the Introductory Section,

'For an isotropic, linear elastic material, thermodynamics requires ν to fall within $-1 \leq \nu \leq 0.5$,¹³ being from the most extendable to the most incompressible, although the limitation can be broken for anisotropic materials.'

In the Discussion Section,

'The microstructure of CMG films is strongly anisotropic, and it should be noted that the aligning process of CPLs only results in in-plane NPR effects, while the Poisson's ratio measured in the out-of-plane direction is always positive, with a reduction in the thickness of CMG films by $\sim 20\%$ at a tensile strain of 3.5% as reported in our previous work.²⁷'

(2) Poisson's ratio is in general not isotropic and hence it is important that when reporting it one also indicates the place where it is being measured and the direction of loading. Thus in this case, the symbol ν in equation (1) should be replaced by ν_{yx} to indicate that the measurement is being done in the xy-plane for stretching in the y-dimension.

Reply: Thank you for the suggestion. The notation was updated.

Changes made:

In the Section Results/Masurement of Poisson's ratios,

$$\nu_{yx} = -d\varepsilon_x / d\varepsilon_y$$

(3) In the measurement of the Poisson's ratio, the same result could probably have been obtained from data of the measurements of dimensions (say 'X' and 'Y') without introducing stress since the instantaneous Poisson's ratio $\nu_{yx} = -(dX/X)/(dY/Y) = -(dX/dY)*(Y/X)$. The reviewer prefers if this form of the equation is also included and the authors check whether the data can be manipulated in such a way that the Poisson's ratio is calculating without invoking a third parameter, in this case stress.

Reply: Thanks for the suggestion, following which the manuscript was rendered.

Changes made:

See our changes made to your previous comment.

(4) The reviewer is pleased to note that finding of this work is confirming earlier work by Grima et al. (ref. 17) which had likened wrinkled graphene films to a macroscopic 'crumpled paper' model. This should be explicitly stated.

Reply: We added discussions on the Grima model and related ones, in both the introductory part and the model analysis section.

Changes made:

In the Introductory Section,

'Grima et al. discovered that the conformation of graphene can be modified via the introduction of defects (by removing atoms), and the crumpled conformation of graphene results in tunable NPRs, where a 'crumpled paper' model was proposed to explain the NPR behaviors of graphene.¹⁸ It is also shown that graphene monolayers exhibit large NPRs in specific directions if defects are introduced in a periodic arrangement, yielding a 'wavy' form of graphene.²²'

In the Section Results/Theoretical Analysis,

'With these arguments, we rationalize the mechanisms through a representative volume element (RVE) featuring wavy textures of CPLs (Fig. 4d), following the spirit of the model of foldable structures proposed by Grima et al.³⁸ and the cubic lattice

model coined by Baughman and co-workers.³⁹

(5) Can the authors comment on isotropy/anisotropy in the Poisson's ratio? Here, the Poisson's ratio is measured for stretching in a particular direction. Whilst it may be inferred that the negative in-plane Poisson's ratio is likely to be accompanied by a large positive Poisson's ratio as the sheet is becoming increasingly more flat as it is stretched, it may equally be extrapolated that the in-plane Poisson's ratio measured for stretching in the Y direction could have been measured for stretching in any other in-plane direction. The authors should also note that since the original Grima et al. paper (ref. 17), there was also another paper which showed that graphene could be made to exhibit even higher negative Poisson's ratio in specific directions if the defects are placed in a periodic manner to get a 'wavy' form of graphene (see Grima et al. (2018) *Ann der Physy.*). This latter paper should also be cited.

Reply: The out-of-plane Poisson's ratio is positive, based on our experimental measurements. We added relevant discussions in the revision. The recommended references were also discussed.

Changes made:

For the discussion on the references, see our changes made to your previous comment.

In the Discussion Section,

'The microstructure of CMG films is strongly anisotropic, and it should be noted that the aligning process of CPLs only results in in-plane NPR effects, while the Poisson's ratio measured in the out-of-plane direction is always positive, with a reduction in the thickness of CMG films by ~20% at a tensile strain of 3.5% as reported in our previous work.²⁷

(6) The authors should specifically that it is 'shape' of the sheet which is giving rise to the specific Poisson's ratio properties, not the actual chemical composition. The authors should check their manuscript well to ensure that this is clearly reflected. In particular, the sentence on the few lines of Page 7 should be re-phrased.

Reply: To clarify the 'shape'-based mechanism of negative Poisson's effects, we introduced the notion of close-packed laminates (CPLs) to capture the microstructural basis. The mis-alignment, wrinkles and delamination of CPLs and their evolution under mechanical loading explain our experimentally observed well-tuned negative Poisson's effects. The relation between chemical composition and the inter-connected

network of CPLs was discussed in text (Page 7 in the previous submission of this work).

Changes made:

The notion of close-packed laminates and related discussions was added to the revised manuscript.

In Section Results/Microstructural basis of the negative Poisson's effects,

‘Considering all of these factors, we conclude that the Poisson’s ratio of CMG films is highly related to the topological structure of CPLs, which can be modulated by the chemical structures of GO building blocks and the film-forming methods. The wrinkled textures of CPLs enhance the NPR effect of the CMG films. After reduction with hydroiodic acid solution, the rGO films feature similar hierarchical structures with wrinkled textures and the auxetic behavior as GO films (Fig. 2c, Supplementary Fig. 2).’

The model part was re-written (see Section Results/Theoretical analysis and the SI).

(7) The reviewer is of the opinion that the mathematical model present (which is a variation of an earlier model presented in 2011 by Grima et al., DOI 10.1002/pssb.201083982) should be presented in the SI. The main text should just give a summary of the model, together with the appropriate reference to earlier work, which apart from this 2011 paper should also cite a much earlier 1998 paper ‘Negative Poisson's ratios as a common feature of cubic metals’ by RH Baughman et al.

Reply: Thank you for your suggestions. We moved the detailed description of our theoretical model into the SI, and the references were discussed in our model analysis section.

Changes made:

See our changes made to your comment #4, and the Supplementary Information for the detailed description of our theoretical model.

(8) The sentence ... indicating that the NPR, i.e., auxetics, is an intrinsic property of CMG films ... should be rephrased ... ‘auxetics’ should be ‘auxeticity’ and the word ‘intrinsic’ is not the best word to use.

Reply: We corrected these issues.

Changes made:

The issues mentioned were corrected and the whole manuscript was further polished.

(9) Towards the end of the Discussion section, within the sentence starting: The findings provide an understanding of the performance... 'additiona' should be 'addition'.

Reply: We corrected this issue.

Changes made:

The issue mentioned was corrected and the whole manuscript was further polished.

Reviewer #2:

In this paper, the authors reported a wide range of tunable negative Poisson's ratio in CMG films in experiments. They found that different synthesis procedures always led to a NPR. They attributed the measured NPR to the intrinsic wrinkles and disordered stacks in CMG films. Via controlling synthesis procedures (such as pre-stretching) to tune the microstructure hierarchy, they can obtain different NPRs in experiments. This feature is very important in practical applications. NPR is an interesting property that has stimulated intensive research interests recently. It has great promises in many applications ranging from aircraft, defense, to biomedical technologies. The authors also proposed a mechanical model that can quantitatively predict the NPR, which agrees with experimental results well. This model will be useful for design. This is an interesting work. This referee has some technical issues and requests the authors to address.

Reply: Thank you for your recommendation. We polished our manuscript following the suggestions from you and other reviewers. Please find below a detailed, point-to-point responses to your minor comments.

(1) The authors showed that the pre-stretching during sample synthesis is an effective way to change NPR. But they did not provide experimental details of how to apply the pre-stretching. This information could be useful for others to further develop this method for real applications.

Reply: Thank you for this suggestion. We updated our manuscript by including the experimental details related to pre-stretching.

Changes made:

In the Section Methods/Preparation of GO and rGO films,

‘Pre-stretched GO films were prepared by sandwiching GO rectangular strips (3 mm × 30 mm) on a home-made apparatus with a given loading under 100% relative humidity (RH) for one week, followed by air-drying the resultant GO films under 50% RH for one day. The GO strips were fabricated by razor blade cutting wet GO films, which were fabricated by cast-drying GO dispersions (5 mg mL⁻¹) on PTFE petri dish under 50% RH for two days.’

(2) Could the authors explain how they estimated the quantity alpha in Eq. (6) from experimental measurements? This quantity is critical in their model.

Reply: α is the angle of alignment of the GO sheets measured in the experiments. We

assumed that the value of $\tan\alpha$ is linear to that of $\tan\theta$, where θ is the angle between the sticks and the basal plane of GO films in our theoretical model, measuring the amplitude of out-of-plane corrugation of the close-packed laminates in the CMG films, resulted from disordered stacking, wrinkles or delamination. We clarify these points in the revised manuscript.

Changes made:

In Section Results/Theoretical analysis,

‘The angle θ between the sticks and the basal plane of a CMG film measures the out-of-plane corrugation of CPLs, and is assumed to be linear to the alignment angle α measured in polarized Raman experiments.’

Remarks were also added to the caption for Fig. 4.

(3) In Fig. S3, the CMG films have nonlinear elasticity. Does this have any correlation with the variation of the instantaneous Poisson ratio as shown in 2b?

Reply: The instantaneous Poisson’s ratios in Fig. 2b (in previous submission) were obtained by piece-wisely fitting the data of ϵ_X - ϵ_Y measured from a single sample. We revised our procedure by introducing statistical averaging over three samples (see revised Fig. 2b, based on data from Fig. S4b). The results, combined with the fact that the relation between transverse and longitudinal strain is almost linear, suggest that the previous variation in measured Poisson’s ratio is statistical, without which the value of Poisson’s ratio is a constant.

Changes made:

Figure 2 was updated.

Figure 2 | Typical mechanical responses of CMG films. (a) The transverse strain ϵ_X plotted as a function of the longitudinal strain ϵ_Y of e -GO₀ films with a tensile loading in the Y direction. **(b)** The instantaneous Poisson's ratios of e -GO₀ films averaged from

three samples in Fig. S4b, plotted with standard deviation. (c) Poisson's ratios of CMG (GO or rGO) films. (d) Poisson's ratios of e -GO₀ films under cyclic uniaxial tensile loading with a peak strain of 1.5%. Source data are provided as a Source Data file.

(4) They have used symbol alpha in Eq. (2) and (6). But apparently, they have a different meaning in these two equations.

Reply: The meaning of α is the same (the description of our theoretical model was now moved to the SI). We assumed that the value of $\tan\alpha$ is linear to that of $\tan\theta$ so the model prediction can be compared with experimental measurement.

Changes made:

See our changes made to your comment #2.

(5) In Figure 1b, the U displacement field is somehow not aligned with the loading direction. Could the authors provide some explanation? Will this affect the calculated strain value?

Reply: Thank you for your critical comments. FDIC is very sensitive to the sample alignment in the calculation of displacement fields. Possible causes could be the imperfect aligning in clipping. Currently we cannot avoid this without damaging the sample. However, for our FDIC data, the angle of misalignment is ~ 5 degree, corresponding to an error in the strain of ~ 0.001 . The FDIC-calculated strain is an order of magnitude of this level of error, and thus we conclude the strain measurement is reliable.

Changes made:

In Section Methods/Mechanical Characterization,

‘It should be noted that the displacement measurement in FDIC is sensitive to the sample alignment, and errors may arise from sample clipping. However, for our FDIC data, the angle of misalignment is ~ 5 degree, corresponding to an error in the strain of ~ 0.001 . The FDIC-calculated strain is an order of magnitude of this level of error, and thus the strain measurement is reliable.’

(6) In page 3, 2nd paragraph, 4th line, the authors stated 'Under an uniaxial tension, all monolithic and homogeneous ...'. Did they refer to isotropic materials?

Reply: Here we refer to ‘most of the common materials’. For isotropic, linear elastic material, thermodynamics requires ν to fall within $-1 \leq \nu \leq 0.5$. We clarified this point

in the revision.

Changes made:

In the Introductory Section,

‘Under uniaxial tension, most of the common materials contract in lateral directions to compensate for stretch-induced volume change, giving positive Poisson’s ratios.’

(7) page 3, first paragraph, line 7, 'stifness' typo?

Reply: We corrected this typo and polished the whole manuscript.

Changes made:

The issue mentioned was corrected and the whole manuscript was further polished.

Reviewer #3:

Wen et al. present a study on Chemically Modified Graphene Films with Tunable Negative Poisson's Ratios. The manuscript is well written but I believe the conclusion are not adequately supported by the data presented and publication in any form would be premature.

Reply: Thank you for your encouraging comments. We improved our work based on your suggestions and provide point-to-point responses to your suggestions. Please see below for details.

1. It is unclear how the authors assume monolithic properties for a GO or rGO microstructure with several nanoscopic platelets stacked. The underlying theoretical assumptions should be clearly explained to place this study on a solid theoretical footing.

Reply: Thanks for this critical comment. We agree that the microstructural basis of our theoretical model was not well described in the previous submission. In this revision, we re-wrote the model part and updated the whole manuscript to make the presentation clear. From the SEM images and our understandings in previous work, we introduced the notion of close-packing laminates (CPLs), the inter-connected network of which is considered as a model description of the CMG films. With this assumption, the wrinkles, delamination of CPLs, as well as their ordered and disordered stacks in CMG films evolve under uniaxial tension. The corrugation of the network of 2D CPLs is reduced during the process, leading to the in-plane negative Poisson's effects. This proposed mechanism is validated by our experimental characterization and theoretical analysis (see below for more detailed description in response to your comments #3, #4 and #6).

Changes made:

In Section Results/Theoretical analysis,

Theoretical analysis. To understand the microstructural evolution of CMG films under tensile loading and quantifying the NPR behaviors, we constructed a multiscale model for the CMG films based on experimental evidences. Specifically, based on SEM and optical images of the cross-section and surface topography of CMG films, respectively (Fig. 4a-b), we illustrated the microstructures of monolithic CMG films in Fig. 4c, which include inter-connected CPLs (or GO multilayers with typical thickness of ~10-30 nm). Their evolution under tensile loading involves aligning of disordered stacks of CPLs, reduction in the out-of-plane corrugation of wrinkled CPLs and their delamination. These processes are activated along the tensile

direction, but the mis-alignment in other directions is also suppressed, as a consequence, endowing the CMG films with NPRs (Fig. 4c). The evidence of these processes can be found from the polarized Raman data. Compared to the flattening process of a GO or graphene monolayer,²⁰ the out-of-plane corrugation of the inter-connected network of CPLs in the CMG film is stabilized by its microstructural hierarchy, and can have much higher amplitude, leading to more prominent NPR effects as we identified experimentally. With these arguments, we rationalize the mechanisms through a representative volume element (RVE) featuring wavy textures of CPLs (Fig. 4d), following the spirit of the model of foldable structures proposed by Grima et al.³⁸ and the cubic lattice model coined by Baughman and co-workers.³⁹ In our model, the sticks (solid and dash lines in Fig. 4d) represent the out-of-plane corrugation of CPLs resulted from disordered stacks, wrinkles, or delamination. The angle θ between the sticks and the basal plane of a CMG film measures the out-of-plane corrugation of CPLs, and is assumed to be linear to the alignment angle α measured in polarized Raman experiments. The ν - α relation predicted from our model (see Supplementary Information for details) align well with experimental data (Fig. 4e), suggesting that the NPR effect of CMG films does originates from the wavy texture of the inter-connected CPL network and will be reduced if the alignment of CPLs improves.'

Details of the theoretical model was moved to the SI.

Figure 4 was updated.

Fig. 4 (a) Wrinkles, delamination of CPLs and their ordered, disordered stacks in e -GO₀ films characterized by SEM. **(b)** 3D reconstruction of the surface topography of

e-GO₀ films through optical microscopy. (c) Illustration of the multiscale model of a CMG film featuring an inter-connected network of CPLs. (d) A 4-stick RVE model, where the red configuration is stretched by force F to the blue one. (e) Theoretical predictions and experimental measurements of the Poisson's ratio as a function of the angle of alignment, α , between GO sheets, which are assumed to be linear to the misalignment between CPLs, θ . Source data are provided as a Source Data file.

2. The authors should explain if there is any relative movement of the flakes in the microstructures as a function of loading? How does this affect the Poisson ratios computed?

Reply: The resolution of our experimental characterization techniques does not allow us to conclude on the relative movements of the constituting sheets or CPLs. We expect that the topology of inter-connected network of CPLs is relatively robust due to the strong topological constraint. If the relative movement between CPLs is significant, we expect that the suppression of CPL corrugation will be weakened, resulting in a reduced NPR effect.

Within the CPLs, the GO sheets (I am not sure if this is the 'flake' you mentioned in the comment) could be under shear or activated to slide as a function of loading. Since the NPR effect is manifested at the CPL network level, we expect that the relative movement at sheet level may not affect the NPRs calculated.

We added relevant discussion in to the text.

Changes made:

In the Discussion Section,

'Our hypothesis on the microstructural evolution in CMG films is limited by the resolution of structural characterization techniques. Relative sliding between CPLs is not considered, which could weaken the suppression of out-of-plane corrugations of CPLs under tensile loading, and thus reduces the NPR effect. The morphological changes of GO sheets in the CPLs are expected to be strongly constrained by the layer-by-layer packing. The chemical structures of GO sheets are thus correlated to the NPR behaviors through their modulation in the film-forming processes.'

3. The authors should explain if there is solvent entrapment within the microstructures that gives rise to differences seen for evaporated vs vacuum filtered and 0 vs 35 fabrication methods.

Reply: Thank you for this comment. To address this issue, we conducted thermal

gravimetric analysis (TGA) to evaluate water (solvent) entrapment within the film as the matrix. As shown in the weight-loss curves shown in Figure R1, all the specimens, irrespective of their GO precursors and film-forming methods, exhibit similar weight loss at low temperature ($< 125\text{ }^{\circ}\text{C}$), which is attributed to the elimination of water molecules (*ACS Nano* **2012**, *6*, 8357). This result suggest that the solvent entrapped within the microstructures does not depend on the fabrication methods.

However, the temperature of decomposition of CMG films (T_d) depends on the GO precursors and film forming methods. It has been reported that the value of T_d is related to gas escaping from the film matrix (*Chem. Mater.* **2012**, *24*, 1276.) The GO films processed from GO_{35} and the filtrated method demonstrate a higher T_d than the others, which is explained by the stronger hydrogen bonding between GO_{35} sheets (through more carboxyl groups) and more ordered structure in the f -GO films by filtration, where the gas escape from the film matrix is suppressed (*Chem. Sci.* **2016**, *7*, 1874.).

Figure R1. TGA data measured for GO films processed from different GO precursors and film forming methods. All measurements were performed under argon atmosphere at a heating rate of $5\text{ }^{\circ}\text{C min}^{-1}$.

Changes made:

In Section Results/Preparation of CMG films,

‘Thermal gravimetric analysis (TGA) shows that the solvent entrapped in the film does not depend on the fabrication methods.’

4. Raman and UV-Vis are microscopic probes and it is unclear how the macroscopic properties (Poisson ratio) of the microstructure are directly related to microscopic properties.

Reply: Thank you for the comments. We agree that Raman and UV-Vis are microscopic probes and provide information on the chemical structures and sheet alignment, which are related to the NPR effect as explained below.

The UV-vis spectra probe the chemical structures GO sheets. GO₀ sheets prepared by low temperature oxidation protocol feature large graphitic domains (*Adv. Mater.* **2013**, *25*, 3583; *Nat. Chem.* **2014**, *6*, 151), which facilitate stronger interaction between GO sheets (*J. Mech. Phys. Solids* **2012**, *60*, 591). As a result, in film-forming processes by evaporation-induced assembly or vacuum-assisted filtration, the mobility of GO₀ sheets in the dispersions is limited, and it becomes difficult to adjust the sheet conformation to adopt an energetically-favorable, ordered alignment, resulting in the formation of GO₀ films, where the hierarchical structures of GO sheet-containing CPLs is less ordered compared with GO₃₅ films.

Polarized Raman spectroscopy assesses the alignment of GO sheets in the CMG films through the G band intensity on polarization geometries according to a “depolarization effect” (*ACS Nano* **2011**, *5*, 2392; *Adv. Funct. Mater.* **2016**, *26*, 7003). Based on the assumption that the inter-connected network of CPLs characterizes the hierarchical structure of CMG films, the polarized Raman data measures the suppression of the out-of-plane corrugation of CPLs upon tensile loading, which results in the NPR effect.

Changes made:

In Section Results/Microstructural basis of the negative Poisson's effects,

‘The structural differences between the GO sheets prepared by different synthetic protocols are further identified by UV-Vis absorption spectra. The main absorption peak of GO around 230 nm is associated with the sp^2 domain in GO sheets. The absorption peak of GO₀ (236 nm) is red-shifted with respect to that of GO₃₅ (231 nm), demonstrating that GO₀ sheets have a larger graphitic domain than that in GO₃₅, since the carbon atom rupture is prevented to some extent by a relatively lower temperature of oxidization (Supplementary Fig. 6).^{33,34} The low defect density and larger sp^2 domains within GO₀ basal planes facilitate the van der Waals interactions between neighboring GO₀ sheets.^{34,35} Consequently, in film-forming processes by evaporation-induced assembly or vacuum-assisted filtration, the mobility of GO₀ sheets in the dispersions is limited, and it becomes difficult to adjust the sheet conformation to adopt an energetically-favorable, ordered alignment, resulting in the formation of GO₀ films, where the hierarchical structures of GO sheet-containing CPLs is less ordered compared with GO₃₅ films.’

‘Polarized Raman spectra were recorded by directing an incident laser beam parallel to the base plane of *e*-GO₀ films with pre-stretch. As shown in Fig. 3c, α is the average angle of alignment between the CPLs and the loading direction, and β is the angle between the electric field vector of the incident laser and the base plane of the *e*-GO₀ film, which is tuned from 0° to 360° (Supplementary Fig. 9). The G band intensity in

polarized Raman spectra of the CMG films is plotted against the angles α and β in Fig. 3c, which follow the equation

$$I_G = \frac{c^2}{2} \cos^2 \alpha \{2 + \cos[2(\alpha - \beta)] + \cos[2(\alpha + \beta)]\} \quad (2)$$

For GO samples with the angle of alignment α , the value of $I_G(\beta)$ characterized experimentally reaches a maximum at $\beta = 0^\circ$ and 180° ($I_G(\parallel)$), whereas $I_G(\beta)$ reaches a minimum at $\beta = 90^\circ$ and 270° ($I_G(\perp)$).³⁷ The ratio $I_G(\parallel) / I_G(\perp)$ equals $\cot^2 \alpha$ (Fig. 3c). Fitting the data of normalized Raman G band intensity with the value of β between 0° and 360° using Equation 2, one obtains the value of α decreasing from 37.56° to 35.84° , and 32.78° upon enhancing pre-stretch (Figs. 3c-d). Based on these results, it becomes clear that the NPR of CMG films can be tuned by modifying their hierarchical microstructures, where the inter-connected network of CPLs define the NPR behaviors of CMG films.'

In Discussion Section,

'The chemical structures of GO sheets are thus correlated to the NPR behaviors through their modulation in the film-forming processes.'

5. How controllable and re-producible is the Poisson ratio for each method of fabrication? Why is the variability largest for the evaporated sample compared to vacuum filtered?

Reply: The Poisson's ratio of CMG films were determined by statistical averaging of three samples. The results show that the standard deviation is minor compared to the averaged Poisson's ratios. As revealed from the XRD patterns, evaporated samples demonstrate a larger interlayer spacing and full-width at half-maximum (FWHM) than the vacuum filtered ones. GO sheets intend to form a low-density dynamic network at a relatively higher concentration ($\geq 3 \text{ mg ml}^{-1}$) due to the presence of strong physical cross-linking sites, leading to formation of CMG films with more wrinkled textures of the CPLs. The more disordered structures of the *e*-GO films explain the larger variability compared to the *f*-GO films.

Changes made:

'As seen in Fig. 3a, with the same GO precursors, evaporation-induced assembly of concentrated GO dispersion (5 mg ml^{-1}) produces *e*-GO films with larger interlayer spacing and FWHM, together with more significant NPR effects than the *f*-GO films fabricated by vacuum-assisted filtration from dilute GO dispersions (1 mg ml^{-1}). GO sheets intend to form a low-density dynamic network at a relatively higher

concentration ($\geq 3 \text{ mg ml}^{-1}$) due to the presence of strong physical cross-linking sites,^{26,36} leading to formation of CMG films with more wrinkled textures of the CPLs.’

6. The authors attribute much of the observed properties to wrinkles and crumples in GO or rGO but present no direct evidence that is conclusive/unambiguous. It is hence unclear what is the fundamental mechanisms by which the microstructures exhibit negative Poisson’s ratio?

Reply: Thank you for the comment. To clarify the fundamental mechanism of NPR effects characterized for the CMG films, we updated the manuscript with the notion of close-packed laminates (CPLs) introduced. The NPR effect observed for the CMG films arises from the suppression of wrinkles, delamination of CPLs, as well as their disordered stacks under tensile loading.

Changes made:

See our changes made in response to your comment #1.

7. How do the authors rule out delamination of the films comprising the microstructure leading to the observed negative Poisson ratio?

Reply: In this revision, we include delamination of the CPLs in our discussion. Instead, we consider together with the disordered stacking, wrinkling, delamination is the structural basis of the misalignment between close-packed laminates, and is the origin of experimentally observed negative Poisson’s ratio. To clarify this, we introduce an illustration of our model used to analyze the microstructural evolution of CMG films under tensile loads (Fig. 4c), and clarify the issues you raised.

Changes made:

See our changes made in response to your comment #1.

REVIEWERS' COMMENTS:

Reviewer #1 (Remarks to the Author):

The authors have addressed all the issues raised by the reviewers in a satisfactory manner and I thus recommend acceptance in its present form. However, I suggest that the model section of the Supp Info is typeset better and that the the derived Poisson's ratio is given the suffixes as in the main text so as to indicate the plane where the Poisson's ratio is being measured and the direction of loading.

Reviewer #2 (Remarks to the Author):

After reading the author's reply and their new manuscript, this referee is satisfied with their replies to comments and the corresponding changes to manuscript, particularly their answer to comment (3). This referee also believed that the update/clarification of their model is necessary. This referee recommends acceptance.

Reviewer #3 (Remarks to the Author):

The authors have revised the manuscript significantly to address most of the comments.

Much the observed NPR is attributed to wrinkles and crumples but contributions from relative movement of the flakes (individual layers of mono/few layer GO), de-lamination of the flakes and solvent entrapment between the flakes remain un-known in the absence of clear experimental evidence. It may be useful to incorporate flake size distributions into the model to observe its influence on the NPR.

A small paragraph in the discussion section outlining the limitations of the approach and any resolution limits of the techniques used is warranted (like the authors have done in the response to reviewers).

REPLIES TO REVIEWER COMMENTS

REVIEWER #1:

The authors have addressed all the issues raised by the reviewers in a satisfactorily manner and I thus recommend acceptance in its present form. However, I suggest that the model section of the Supp Info is typeset better and that the derived Poisson's ratio is given the suffixes as in the main text so as to indicate the plane where the Poisson's ratio is being measured and the direction of loading.

Reply: Thank you for the recommendation, and we update Supplementary Note 4 (the theoretical model) following your suggestion.

Changes made:

In Supplementary Note 4

$$v_{yx} = -\frac{\Delta w/w}{\Delta l/l} = \frac{(1 - \frac{s^2 K}{3D}) \tan^2 \theta}{1 + \frac{s^2 K}{3D} \tan^2 \theta} \quad (\text{S3})$$

by assuming $w = l$ with the assumption of in-plane isotropy in the CMG films. We also assume that $\tan \alpha$ is a linearly function of $\tan \theta$ with a pre-factor N ($\tan \theta = h/l = N \tan \alpha$).

Finally, denote $s^2 K/3D$ as M , the Poisson's ratio can be expressed as

$$v_{yx} = \frac{(1-M)N^2 \tan^2 \alpha}{1+MN^2 \tan^2 \alpha} \quad (\text{S4})'$$

REVIEWER #2:

After reading the author's reply and their new manuscript, this referee is satisfied with their replies to comments and the corresponding changes to manuscript, particularly their answer to comment (3). This referee also believed that the update/clarification of their model is necessary. This referee recommends acceptance.

Reply: Thank you for your recommendation.

REVIEWER #3:

The authors have revised the manuscript significantly to address most of the comments.

Much the observed NPR is attributed to wrinkles and crumples but contributions from relative movement of the flakes (individual layers of mono/few layer GO), de-lamination of the flakes and solvent entrapment between the flakes remain un-known in the absence of clear experimental evidence. It may be useful to incorporate flake size distributions into the model to observe its influence on the NPR. A small paragraph in the discussion section outlining the limitations of the approach and any resolution limits of the techniques used is warranted (like the authors have done in the response to reviewers).

Reply: Thank you for the suggestions. We agree that the distribution of microstructural features of the CMG films can be wide due to the coarse control of film preparation process, and thus our RVE-based model is limited by neglecting the variation. However, we believe this simplified model captures the most essential mechanism by introducing the concept of CPLs, and when we have more reliable statistics of the microstructural parameters from experimental measurements, we can extend our theoretical analysis into numerical models that can take into account for the details.

Changes made:

In Discussion,

‘Our hypothesis on the microstructural evolution in CMG films is limited by the resolution of structural characterization techniques. Relative sliding between CPLs is not considered, which may weaken the suppression of out-of-plane corrugations of CPLs under tensile loading, and thus reduces the NPR effect. The morphological changes of GO sheets in the CPLs are expected to be strongly constrained by the layer-by-layer packing. The chemical structures of GO sheets are thus correlated to the NPR behaviors through their modulation in the film-forming processes.

Future work on the characteristic microstructural features and their statistics from

experimental measurements is acquired to extract the key parameters that can be used to construct numerical models, which can extend the current analysis by considering more realistic models. Our RVE-based theory neglects the wide distribution of the microstructural features resulted from the coarse control of film preparation processes, but captures the most essential mechanism of the NPR effect by introducing the concept of CPLs. Our experimental characterization and analysis indicate that the wavy textures of close-packed GO laminates as the building blocks are responsible for the observed auxetic behavior in the macroscopic GO and rGO films. The argument is validated by further engineering control of the NPR behaviors. The wrinkled texture of the inter-connected CPL network within the films is readily tailored by the chemical structures of the GO precursors and the film-forming methods, resulting in films with tunable NPR values.'